# PIE: PSEUDO-INVERTIBLE ENCODER

## ABSTRACT

We consider the problem of information compression from high dimensional data. Where many studies consider the problem of compression by non-invertible transformations, we emphasize the importance of invertible compression. We introduce new class of likelihood-based Auto-Encoders with pseudo bijective architecture, which we call Pseudo Invertible Encoders. We provide the theoretical explanation of their principles. We evaluate Gaussian Pseudo Invertible Encoder on MNIST, where our model outperforms WAE and VAE in sharpness of the generated images.

## 1 INTRODUCTION

We consider the problem of information compression from high dimensional data. Where many studies consider the problem of compression by non-invertible transformations, we emphasize the importance of invertible compression as there are many cases where one cannot or will not decide a priori what part of the information is important and what part is not. Compression of images for person ID in a small company requires less resolution then person ID at an airport. To loose part of the information without harm to the future purpose of viewing the picture requires knowing the purpose upfront. Therefore, the fundamental advantage of invertible information compression is that compression can be undone if a future purpose so requires.

Recent advances of classification models have demonstrated that deep learning architectures of proper design do not lead to information loss while still being able to achieve state-of-the-art in classification performance. These *i*-RevNet models Jacobsen et al. (2018) implement a small but essential modification of the popular RevNet models while achieving invertibility and a performance similar to the standard RevNet Gomez et al. (2017). This is of great interest as it contradicts the intuition that information loss is essential to achieve good performance in classification Tishby & Zaslavsky (2015). Despite the requirement of the invertibility, flow-based generating models Dinh et al. (2014; 2016); Rezende & Mohamed (2015); Kingma & Dhariwal (2018) demonstrate that the combination of bijective mappings allows one to transform the raw distribution of the input data to any desired distribution and perform the manipulation of the data.

On the other hand, Auto-Encoders have provided the ideal mechanism to reduce the data to the bare minimum while retaining all essential information for a specific task, the one implemented in the loss function. Variational Auto Encoders (VAE) Kingma & Welling (2013) and Wasserstein Auto Encoders (WAE) Tolstikhin et al. (2018) are performing best. They provide an approach for stable training of autoencoders, which demonstrate good results at reconstruction and generation. However, both of these methods involve the optimization of the objective defined on the pixel level. We would emphasise the importance of avoiding the separate decoder part and training the model without relying on the reconstuction quality directly.

Combining the best of Invertible mappings and Auto-Encoders, we introduce Pseudo Invertible Encoder. Our model combines bijectives with restriction and extension of the mappings to the dependent sub-manifolds Fig. 1. The main contributions of this paper are the following:

- We introduce new class of likelihood-based Auto-Encoders, which we call Pseudo Invertible Encoders. We provide the theoretical explanation of their principles.
- We demonstrate the properties of Gaussian Pseudo Invertible Encoder in manifold learning.
- We compare our model with WAE and VAE on MNIST, and report that the sharpness of the images, generated by our models is better.

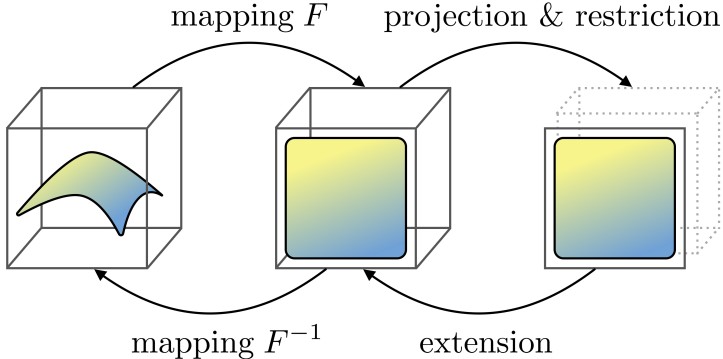

Figure 1: Schematic representation of the proposed mechanism of dimensionality reduction.

## 2 RELATED WORK

### 2.1 INVERTIBLE MODELS

ResNets He et al. (2016) enable Networks to grow even more and thus memory consumption becomes a bottleneck. Gomez et al. (2017) propose a Reversible Residual Network (RevNet) where each layer's activations can be reconstructed from the activations of the next layer. By replacing the residual blocks with coupling layers, they mimic the behaviour of residual blocks while being able to retrieve the original input of the layer. RevNet replaces the residual blocks of ResNets, but also accommodates non-invertible components to train more efficiently. By adding a downsampling operator to the coupling layer, $i$-RevNet circumvents these non-invertible modules (Jacobsen et al., 2018). With this they show that losing information is not a necessary condition to learn representations that generalize well on complicated problems. Although $i$-RevNet circumvents non-invertible modules, data is not compressed and the model is only invertible up to the last layer. All their methods do not allow dimensionality reduction. In current research we build a pseudo invertible model which performs dimensionality reduction.

### 2.2 AUTOENCODERS

Auto-Encoders were first introduced by Rummerhart (1986) as an unsupervised learning algorithm. They are now widely used as a technique for dimension reduction by compressing input data. By training an encoder and a decoder network, and measuring the distance between original and reconstructed data, data can be represented in a latent space. This latent space can then be used for supervised learning algorithms. Instead of learning a compressed representation of the input data Kingma & Welling (2013) propose to learn the parameters of a probability distribution that represent the data. tol introduced new class of models - Wasserstein Auto Encoders, which use Optimal Transport to be trained. These methods require the optimization of the objective function which includes the terms defined on pixel level. Our model does not require such optimization. Moreover, it only perform encoding at training time.

## 3 THEORY

Here we introduce the approach for obtaining dimensionality reduction invertible mappings. Our method is based on the *restriction* of the mappings to low-dimensional manifolds, and *extension* of the inverse mappings with certain constraints (Fig. 2).

### 3.1 RESTRICTION-EXTENSION APPROACH

Given data $\mathbf{x}_i \in \mathcal{X} \subset \mathbb{R}^D$. Assuming that $\mathcal{X}$ is a $d$-dimensional manifold, with $d < D$, we seek to find a mapping $G : \mathbb{R}^D \to \mathbb{R}^d$ invertible on $\mathcal{X}$. In other words, we are looking for a pair of

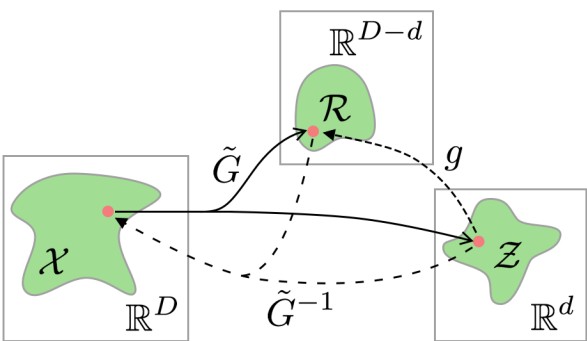

Figure 2: The schematic representation of the *Restriction-Extension* approach. The invertible mapping $\mathcal{X} \leftrightarrow \mathcal{Z}$ is preformed by using the dependent sub-manifold $\mathcal{R} = g(\mathcal{Z})$ and a pair extended functions $\tilde{G}$, $\tilde{G}^{-1}$.

associated functions $G$ and $G^{-1}$ such that

$$
\begin{cases}
G(\mathcal{X}) = \mathcal{Z} \subset \mathbb{R}^d \\
G^{-1}(\mathcal{Z}) = \mathcal{X}
\end{cases}
\tag{1}
$$

Let $\mathcal{R}$ be an open set in $\mathbb{R}^{D-d}$. We use this residual manifold in order to match the dimensionalities of the hidden and initial spaces. Here we introduce the function $g : \mathbb{R}^d \to \mathbb{R}^{D-d}$. With no loss of generality we can say that $\mathcal{R} = g(\mathcal{Z})$. We use the pair of extended functions $\tilde{G} : \mathbb{R}^D \to \mathbb{R}^d \times \mathbb{R}^{D-d}$ and $\tilde{G}^{-1} : \mathbb{R}^d \times \mathbb{R}^{D-d} \to \mathbb{R}^D$ to rewrite Eq. 1:

$$
\begin{cases}
\tilde{G}(\mathcal{X}) = \mathcal{Z} \times \mathcal{R} \\
\tilde{G}^{-1}(\mathcal{Z} \times \mathcal{R}) = \mathcal{X}
\end{cases}
\tag{2}
$$

Rather than searching for the invertible dimensionality reduction mapping directly, we seek to find $\tilde{G}$, the invertible transformation with certain constraints, expressed by $\mathcal{R}$.

In search for $\tilde{G}$, we focus on $F_{\boldsymbol{\theta}} : \mathbb{R}^D \to \mathbb{R}^D$, $F_{\boldsymbol{\theta}} \in \mathcal{F}$, where $\mathcal{F}$ is a parametric family of functions invertible on $\mathbb{R}^D$. We select the function $F_{\boldsymbol{\theta}}$ with parameters $\boldsymbol{\theta}$ which satisfy the constraint:

$$
F_{\boldsymbol{\theta}}^{-1} \circ P_{\mathbb{R}^d \times \mathcal{R}} \circ F_{\boldsymbol{\theta}} = \mathrm{id}_{\mathcal{X}}
\tag{3}
$$

where $P_{\mathbb{R}^d \times \mathcal{R}}$ is the orthogonal projection from $\mathbb{R}^d \times \mathbb{R}^{D-d}$ to $\mathbb{R}^d \times \mathcal{R}$.

Taking into account constraint 3, we derive $F_{\boldsymbol{\theta}}(\mathbf{x}) = [\mathbf{z}, \mathbf{r}]$, where $\mathbf{z} \in \mathcal{Z}$ and $\mathbf{r} \in \mathcal{R}$. By combining this with Eq. 2 we have the desired pair of functions:

$$
\begin{cases}
G(\mathbf{x}) = \mathbf{z}, \\
G^{-1}(\mathbf{z}) = F_{\boldsymbol{\theta}}^{-1}([\mathbf{z}, g(\mathbf{z})])
\end{cases}
\tag{4}
$$

The obtained function $G$ is *Pseudo Invertible Endocer*, or shortly *PIE*.

### 3.2 Log Likelihood Maximization

As we are interested in high dimensional data such as images, the explicit choice of parameters $\boldsymbol{\theta}$ is impossible. We choose $\boldsymbol{\theta}^*$ as a maximizer of the log likelihood of the observed data given the prior $p_{\theta}(\mathbf{x})$:

$$
\boldsymbol{\theta}^* = \arg \max_{\boldsymbol{\theta}} [\log p_{\boldsymbol{\theta}}(\mathbf{x})]
\tag{5}
$$

After a change of variables according to Eq. 4 we obtain

$$
p(\mathbf{x}) = p(F_{\boldsymbol{\theta}}(\mathbf{x})) \left| \det \left( \frac{\partial F_{\boldsymbol{\theta}}}{\partial \mathbf{x}^T} \right) \right|
\tag{6}
$$

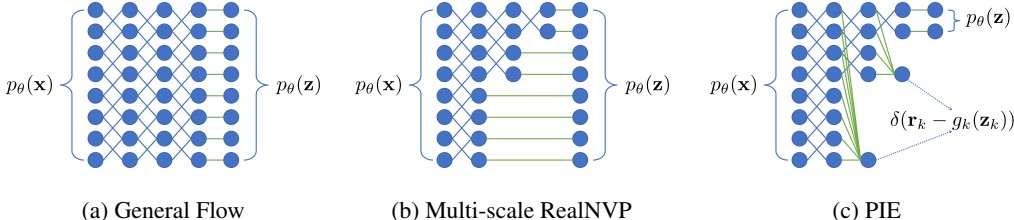

(a) General Flow  (b) Multi-scale RealNVP  (c) PIE

Figure 3: Schematic representation of three types of bijective mappings currently used in normalizing flows. The circles represent the variables. The basic invertible mappings are depicted with blue edges. Green edges represent the aggregation of the variables in objective function. In general normalizing flow (a) all the variables are mapped in the same manner and are propagated through the same number of flows. The multi-scale architecture used in RealNVP (b) transform different variables with different number of flows and afterwards map them to the same distribution. Our model (c) progressively discards part of the variables by hardly constraining their distributions.

Taking into account the constraint 3 we derive the joint distribution for $F_{\boldsymbol{\theta}}(\mathbf{x}) = [\mathbf{z}, \mathbf{r}]$

$$p(F_{\boldsymbol{\theta}}(\mathbf{x})) = p(\mathbf{z}, \mathbf{r}) = p(\mathbf{r}|\mathbf{z})p(\mathbf{z}) \tag{7}$$

$$\int_{\mathcal{X}} p(F_{\boldsymbol{\theta}}(\mathbf{x}))d\mathbf{x} = \int_{\mathcal{R}=g(\mathcal{Z})} \int_{\mathcal{Z}} p(\mathbf{r}|\mathbf{z})p(\mathbf{z})d\mathbf{r}d\mathbf{z} = \int_{\mathcal{R}} \int_{\mathcal{Z}} \delta(\mathbf{r} - g(\mathbf{z}))p(\mathbf{z})d\mathbf{r}d\mathbf{z} \tag{8}$$

$$p(F_{\boldsymbol{\theta}}(\mathbf{x})) = \delta(\mathbf{r} - g(\mathbf{z}))p(\mathbf{z}) \tag{9}$$

Dirac's delta function can be viewed as a limit of sequence of Gaussians:

$$\delta(\mathbf{x}) = \lim_{\epsilon \to 0} \mathcal{N}(\mathbf{x}|\mathbf{0}, \epsilon^2 \mathbf{I}) \tag{10}$$

Let us fix $\epsilon^2 = \epsilon_0^2 \ll 1$. Then

$$\delta(\mathbf{x}) \approx \mathcal{N}(\mathbf{x}|\mathbf{0}, \epsilon_0^2 \mathbf{I}) \tag{11}$$

$$\delta(\mathbf{r} - g(\mathbf{z})) \approx \mathcal{N}(\mathbf{r}|g(\mathbf{z}), \epsilon_0^2 \mathbf{I}) \tag{12}$$

Finally, for the log likelihood we have:

$$\log p(\mathbf{x}) \approx \log p(\mathbf{z}) + \log \mathcal{N}(\mathbf{r}|g(\mathbf{z}), \epsilon_0^2 \mathbf{I}) + \log \left| \det \left( \frac{\partial F_{\boldsymbol{\theta}}}{\partial \mathbf{x}^T} \right) \right| \tag{13}$$

We choose prior distribution $p(\mathbf{z})$ as Standard Gaussian. We search for the parameters by using Gradient Descent.

### 3.3 Composition of Bijectives

The method relies on the function $F_{\boldsymbol{\theta}}$. This choice is challenging by itself. The currently known classes of real-value bijectives are limited. To overcome this issue, we approximate $F_{\boldsymbol{\theta}}$ with a composition of basic bijectives from certain classes:

$$F = F_K \circ F_{k-1} \circ \ldots \circ F_2 \circ F_1 \tag{14}$$

where $F_j = F_j(\cdot|\boldsymbol{\theta}_j) \in \mathcal{F}_j$, $j = 1 \ldots K$.

Taking into account that a composition of *PIE* is also *PIE*, we create a final dimensionality reduction mapping from a sequence of *PIE*s:

$$\mathcal{X} \leftrightarrow \mathcal{Y}_1 \leftrightarrow \mathcal{Y}_2 \leftrightarrow \cdots \leftrightarrow \mathcal{Y}_L \leftrightarrow \mathcal{Z}_1 \tag{15}$$

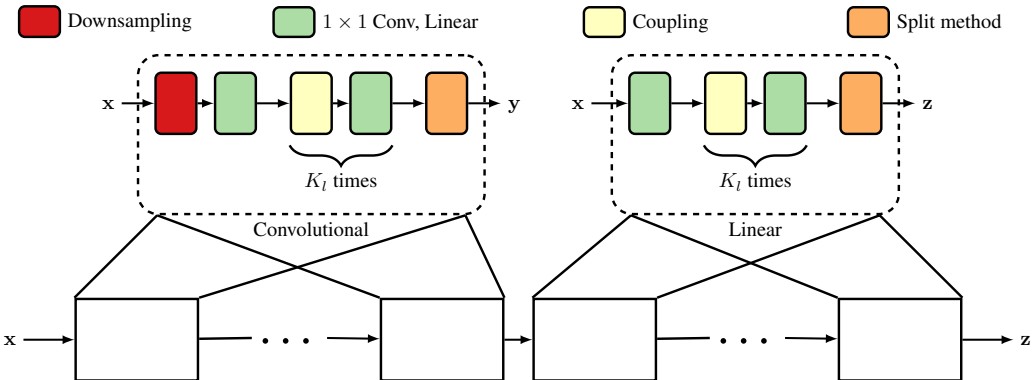

Figure 4: Architecture of the Pseudo-Invertible Encoder. *PIE* consists of convolutional and linear blocks which can be repeated multiple times, as denoted by the three dots between the block structure at the bottom. Each block has $K_l$ repetitions of coupling layers and $1 \times 1$ convolutions.

such that

$$D > \dim \mathcal{Y}_1 > \dim \mathcal{Y}_2 > \ldots > \dim \mathcal{Y}_L > d \qquad (16)$$

where $L < D - d$.

Then the log likelihood is represented as

$$\log p(\mathbf{x}) \approx \log p(\mathbf{z}) + \sum_{l=1}^{L} \log \mathcal{N}(\mathbf{r}_l | g_l(\mathbf{z}_l), \epsilon_0^2 \mathbf{I}) + \sum_{l=1}^{L} \sum_{k=1}^{K_l} \log |\det(\mathbf{J}_{kl})| \qquad (17)$$

where $\mathbf{J}_{kl}$ is the Jacobian of the $k$-th function of the $l$-th *PIE*. The approximation error here depends only on $\epsilon$, according to the Eq. 10. For the simplicity we will now refer to the whole model as *PIE*. The building blocks of this model are *PIE* blocks.

### 3.4 Relation to Normalizing Flows

If we choose the distribution $p(\mathbf{z})$ in Eq. 17 as Standard Gaussian, $g_l(\cdot) = \mathbf{0}$, $\forall l$ and $\epsilon_0 = 1$, then the model can be viewed as Normalizing Flow with multi-scale architecture Dinh et al. (2016) Fig. 3. It was demonstrated in Dinh et al. (2016) that the model with such architecture achieves semantic compression.

## 4 Pseudo-Invertible Encoder

This section introduces the basic bijectives for the Pseudo-Invertible Encoder (*PIE*). We explain what each building bijective consists of and how it fits in the global architecture as shown in Fig. 4.

### 4.1 Architecture

*PIE* is composed of a series of convolutional blocks followed by linear blocks, as depicted in Fig. 4. The convolutional *PIE* blocks consist of series of coupling layers and $1 \times 1$ convolutions. We perform invertible downsampling of the image at the beginning of the convolutional block, by reducing the spatial resolution and increasing the number of channels, keeping the overall number of the variables the same. At the end of the convolutional *PIE* block, the split of variables is performed. One part of the variables is projected to the residual manifold $\mathcal{R}$ while others is feed to the next block. The linear *PIE* blocks are constructed in the same manner. However, the downsampling is not performed and $1 \times 1$ convolutions are replaced invertible linear mappings.

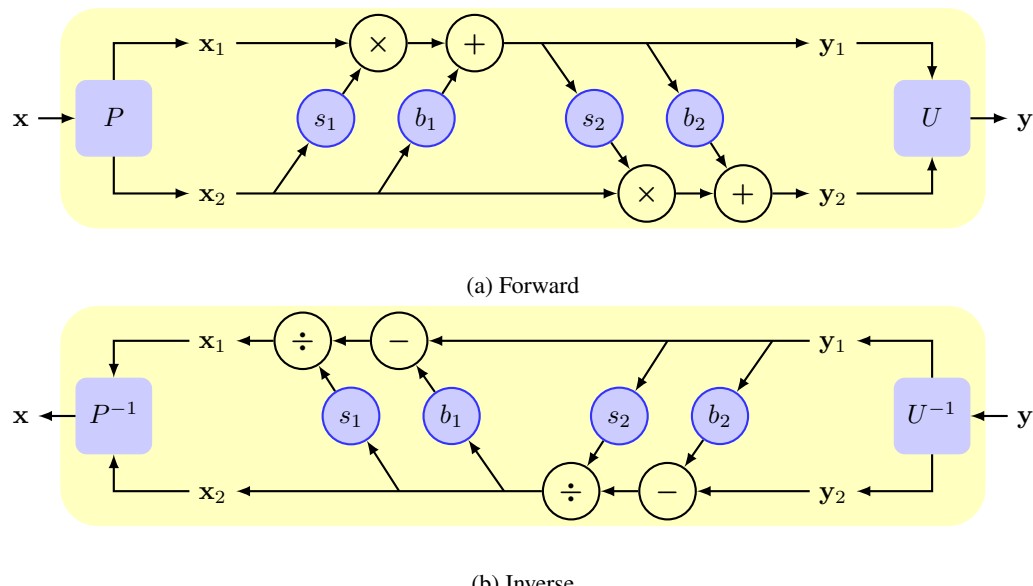

(a) Forward

(b) Inverse

Figure 5: Structure of a coupling block. $P$ partitions the input into two groups of equal length. $U$ unites these group together. In the inverse $P^{-1}$ and $U^{-1}$ are the reverse of these operations respectively.

## 4.2 COUPLING LAYER

In order to enhance the flexibility of the model, we utilize affine coupling layers Fig. 5. We modify the version, introduced in Dinh et al. (2016).

Given input data $\mathbf{x}$, the output $\mathbf{y}$ is obtained by using the mapping:

$$\begin{cases} \mathbf{y}_1 = s_1(\mathbf{x}_2) \odot \mathbf{x}_1 + b_1(\mathbf{x}_2) \\ \mathbf{y}_2 = s_2(\mathbf{y}_1) \odot \mathbf{x}_2 + b_2(\mathbf{y}_1) \end{cases} \Longleftrightarrow \begin{cases} \mathbf{x}_2 = (\mathbf{y}_2 - b_2(\mathbf{y}_1))/s_2(\mathbf{y}_1) \\ \mathbf{x}_1 = (\mathbf{y}_1 - b_1(\mathbf{x}_2))/s_1(\mathbf{x}_2) \end{cases} \tag{18}$$

Here multiplication $\odot$ and division are performed element-wise. The scalings $s_1, s_2$ and the biases $b_1, b_2$ are the functions, parametrized with neural networks. The invertibility is not required for this functions. $\mathbf{x}_1, \mathbf{x}_2$ are the non-intersecting partitions of $\mathbf{x}$. For convolutional blocks we partition the tensors by splitting them into halves along the channels. In case of the linear blocks, we just split the features into halves.

The log determinant of the Jacobian of coupling layer is given by:

$$\log \left| \det \left( \frac{\partial F_{\boldsymbol{\theta}}}{\partial \mathbf{x}^T} \right) \right| = \text{sum}(\log |s_1|) + \text{sum}(\log |s_2|)$$

where $\log |\cdot|$ is calculated element-wise.

## 4.3 INVERTIBLE $1 \times 1$ CONVOLUTION AND LINEAR TRANSFORMATION

The affine couplings operate on non-intersecting parts of the tensor. In order to capture the various correlations between channels and features, the different mechanism of channel permutations were proposed. Kingma & Dhariwal (2018) demonstrated that invertible $1\times1$ convolutions perform better than fixed permutations and reversing of the order of channels Dinh et al. (2016).

We parametrize Invertible $1 \times 1$ Convolutions and invertible linear mappings with Householder Matrices Householder (1958). Given the vector $\mathbf{v}$, the Householder Matrix is computed as:

$$\mathbf{H}(\mathbf{v}) = \mathbf{I} - 2\frac{\mathbf{v}\mathbf{v}^T}{\mathbf{v}^T\mathbf{v}} \tag{19}$$

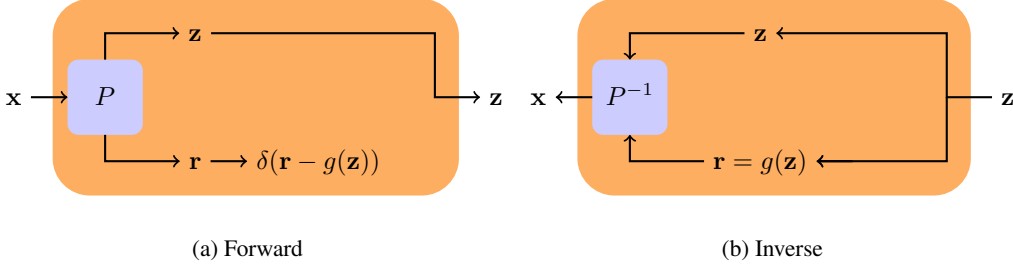

(a) Forward    (b) Inverse

Figure 6: Structure of the split method. $P$ partitions the input into two sub samples. $P^{-1}$ unites these sub samples together.

The obtained matrix is orthogonal. Therefore, its inverse is just its transpose, which makes the computation of the inverse easier comparing to Kingma & Dhariwal (2018). The log determinant of the Jacobian of such transformation is equal to 0.

### 4.4 DOWNSAMPLING

We use invertible downsampling to progressively reduce the spatial size of the tensor and increase the number of its channels. The downsampling with the checkerboard patterns Jacobsen et al. (2018); Dinh et al. (2016) transforms the tensor of size $C \times H \times W$ into a tensor of size $4C \times \frac{H}{2} \times \frac{W}{2}$, where $H, W$ are the height and the width of the image, and $C$ is the number of the channels. The log determinant of the Jacobian of Downsampling is 0 as it just performs permutation.

### 4.5 SPLIT

All the discussed blocks transform the data while preserving its dimensionality. Here we introduce Split block Fig. 6, which is responsible for the *projection*, *restrictions* and *extension*, described in Section 3. It reduces the dimensionality of the data by splitting the variables into two non-intersecting parts $\mathbf{z}, \mathbf{r}$ of dimensionalities $d$ and $D - d$, respectively. $\mathbf{z}$ is kept and is to be processed by the subsequent blocks. $\mathbf{r}$ is constrained to match $\mathcal{N}(\mathbf{r}|g(\mathbf{z}), \epsilon_0^2 \mathbf{I})$. The mappings is defined as

$$\begin{cases} \mathbf{z} = \mathbf{x}|_{\mathbb{R}^d} \\ \mathbf{r} \to \mathcal{N}(\mathbf{r}|g(\mathbf{z}), \epsilon_0^2 \mathbf{I}) \end{cases} \iff \mathbf{x} = [\mathbf{z}, g(\mathbf{z})] \tag{20}$$

## 5 EXPERIMENTS

### 5.1 MANIFOLD LEARNING

For this experiment we trained a Gaussian *PIE* on the MNIST digits dataset. We build *PIE* with 2 convolutional blocks, each splitting the data in the last layer to 50% of the input size. Next we add three linear blocks to *PIE*, reducing the dimensions to 64, 10 and the last block does not reduce the dimensions any further. For each affine transformation we use the three biggest possible Householder reflections. For this experiment we set $K_l$ equal to 3. Optimization is done with the Adam optimizer Kingma & Ba (2014). The model diminishes the number of dimensions from $\mathbb{R}^{784}$ to $\mathbb{R}^{10}$.

This experiment shows the ability of *PIE* to learn a manifold with three different constraints; $\epsilon^2 = 0.01$, $\epsilon^2 = 0.1$ and $\epsilon^2 = 1.0$. The results are shown in Fig. 7. As the constraint gets to loose, as shown in the right column, the model is not able to reconstruct anymore (Fig. 7a). Lower values for $\epsilon^2$ perform better in terms of reconstruction. Too low values, however, sample fuzzy images (Fig. 7b). Narrowing down the distribution to sample from increases the models probability to produce accurate images. This is shown in Fig. 7c where samples are taken from $\mathcal{N}(0, 0.5)$. For both $\epsilon^2 = 0.01$ and $\epsilon^2 = 0.1$ reconstructed images are more accurate.

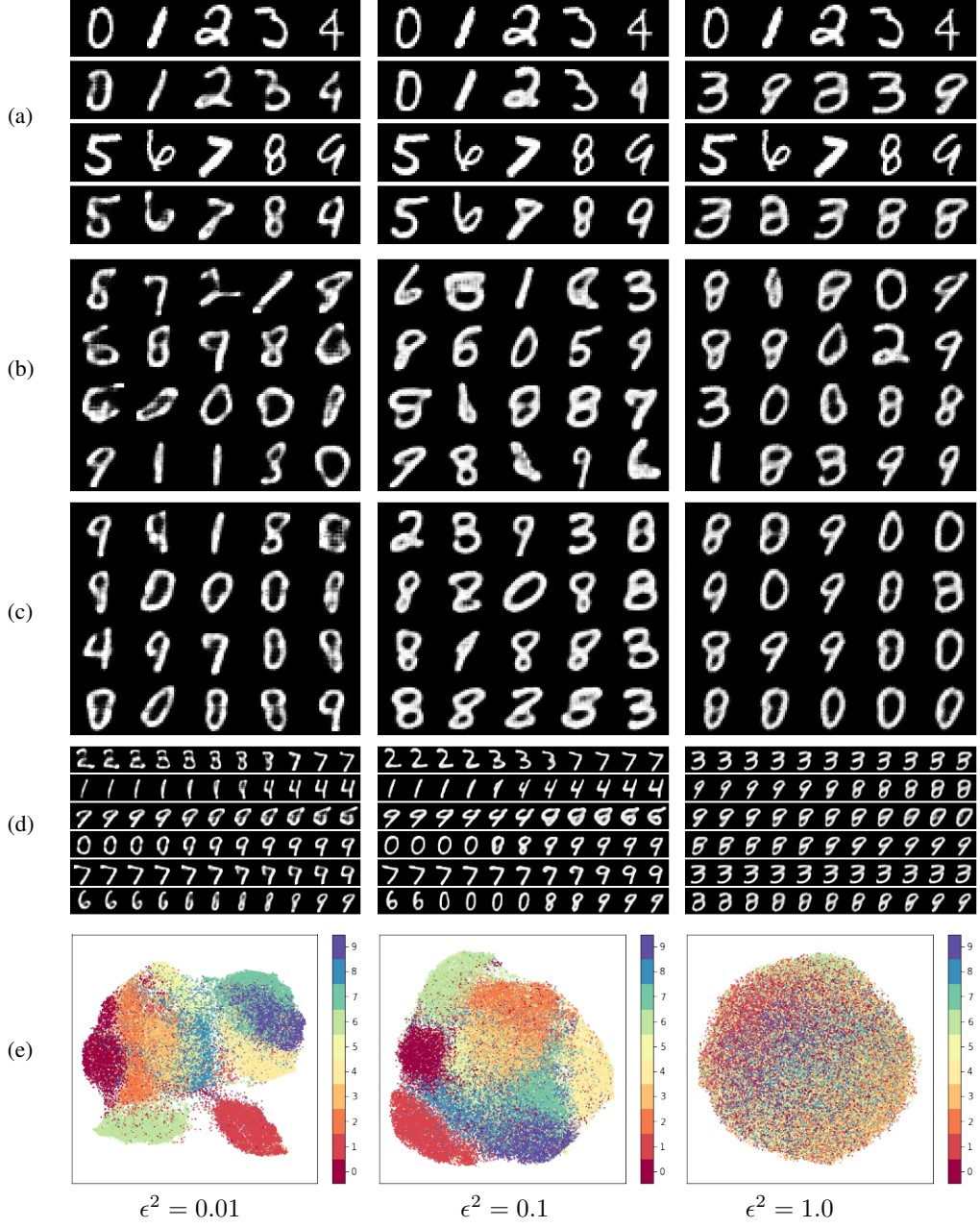

Figure 7: Experiment on MNIST dataset with $\dim(\mathbf{z}) = 10$. (a) shows reconstructions on test data. Row 1 and 3 are original images, row 2 and 4 are reconstructions from $z$-space. (b) and (c) both show reconstruction of a samples $z$-space. (b) is sampled from $\mathcal{N}(0, 1)$, (c) is sampled from $\mathcal{N}(0, 0.5)$. (d) is a linear interpolation between a picture on the left and the right of the image. All digits shown are reconstructed from $z$-space. At last (e) shows UMAP from $\dim(\mathbf{z}) = 10$.

|  | Sharpness |
|---|---|
| True | 0.18 |
| VAE | 0.08 |
| WAE | 0.07 |
| *PIE* | **0.49** |

Table 1: Results for experiment on sharpness on three different models and original images. For all three models a sample of 8 dimensions was taken. The generated images where convolved with Laplace filter and then the variance of activations was averaged over 10000 samples images. Higher values are better.

Fig. 7d shows for each model the linear interpolation from one latent space to another. Both lower values of $\epsilon^2$ (0.01, 0.1) show digits that are quite accurate. When the constraint is loosened to $\epsilon = 1.0$ the interpolation is unable to show distinct values.

This experiment shows that tightening the constraint by decreasing $\epsilon^2$ increases the power of the manifold learned by the model. This is shown again in Fig. 7e where we diminished the number of dimensions even further from $\mathbb{R}^{10}$ to $\mathbb{R}^2$ utilizing UMAP (McInnes et al., 2018). With $\epsilon^2 = 1.0$ UMAP created a manifold with a good Gaussian distribution. However, from the manifold created by *PIE* it was not able to separate distinct digits from each other. Tightening the constraint with a lower $\epsilon^2$ moves the manifold created by UMAP further away from a Gaussian distribution, while it is better able to separate classes from each other.

## 5.2 IMAGE SHARPNESS

It is a well-known problem in VAEs that generated images are smoothened. WAE Tolstikhin et al. (2018) improves over VAEs by utilizing Wasserstein distance function. To test the sharpness of generated images we convolve the grey-scaled images with the Laplace filter. This filter acts as an edge detector. We compute the variance of the activations and average them over 10000 sampled images. If an image is blurry, it means there are less edges and thus more activations will be close to zero, leading to a smaller variance. In this experiment we compare the sharpness of images generated by *PIE* with WAE, VAE and the sharpeness of the original images. For VAE and WAE we take the architecture as described in Radford et al. (2015). For *PIE* we take the architecture as described in section 5.1.

Table 1 shows the results for this experiment. *PIE* outperforms both VAE and WAE in terms of sharpeness of generated images. Images generated by *PIE* are even more sharp then original images from the MNIST dataset. An explanation for this is the use of a checkerboard pattern in the down-sampling layer of the *PIE* convolutional block. With this technique we capture intrinsic properties of the data and are thus able to reconstruct sharper images.

## 6 CONCLUSION

In this paper we have proposed the new class of Auto Encoders, which we call Pseudo Invertible Encoder. We provided a theory which bridges the gap between Auto Encoders and Normalizing Flows. The experiments demonstrate that the proposed model learns the manifold structure and generates sharp images.

## 7 ACKNOWLEDGEMENTS

One of the authors is sponsored by STW project "Imagine: in search for the unknown".

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
