# OpenReview forum: "PIE: Pseudo-Invertible Encoder"
_ICLR.cc/2019/Conference_

### Official Review · AnonReviewer3 · 2018-11-02
**A nice paper but needs stronger experiment results**

**Rating:** 5
**Confidence:** 4

**Review:**

General:
In general, this is a well-written paper and I feel pleasant to read the paper. The paper proposed a model named Pseudo Invertible Autoencoder(PIE) which combines invertible architecture and inference model.

Strength:
1. The explanation of the paper is very clear and consistent.
2. The idea is interesting. A lot of papers related to the inverse problem focus on perfect invertibility, but the author(s) emphasize the importance of invertible compression and relate PIE to the inference model.

Possible Improvements:
1. The experiments could have been more convincing: 1) The only competitors are VAE and WAE. 2)The only data set has been tested was MNIST data set. There are many great works mentioned in the paper and those works should also be compared in a way.
2. The content could be more compact so that more experiments can be shown to support the paper. It seems to me there is too much explanation to previous works in the paper.
3. The paper has 9 pages which exceed the suggestion a little bit.
4. I am not sure if the author(s) checked the grammar of the paper carefully. I found quite few typos in the paper. Page 3: 'Rather then' should be 'Rather than' and 'As we are interested' should be as 'As we are interested in'; Page 4: 'Can me' should be 'Can be'; Page 6: 'Better then' should be 'Better than'; Fig.6 (b): Should it be '0' or 'g(z)'?

Conclusion:
This is a good and clean paper in general. It explains the related work and presents PIE with necessary details. My biggest concern is that empirical validation(experiment) is poor. As a conclusion, I tend to vote for weak rejection.

Minor Suggestion:
Refer to the conference instead of arXiv if the paper was already published.

---

> ### Author Response · Authors · 2018-11-25
> **Reply to  AnonReviewer3**
>
> Thank you for your review!
>
> We have changed Fig. 6 (b) so now it is clear.
> It is g(z) instead of 0. Fig. 6 (b) exactly matches Eq. 20.
>
> We have also fixed the typos in the text.
> We agree that the experimental part is limited.
> We will conduct the experiments on other datasets during the next revision.

---

### Official Review · AnonReviewer2 · 2018-11-04
**An interesting model without thorough evaluations**

**Rating:** 5
**Confidence:** 5

**Review:**

In this paper, an invertible encoding method is proposed for learning latent representations and deep generators via inverting the encoder. The proposed method can be seen as an autoencoder without the need to learn the decoder. This can be computed by inverting the encoder. To the best of my knowledge the proposed method is novel and its building blocks are described adequately.

My main questions are the following: What is the main advantage of this model? Does it make the problem of deep generative model learning tractable? If so, under what conditions?

Discussion of prior art and relevant methods is limited in the paper and it can be extended. The authors may want to consider discussing relevant work on invertible autoencoders (e.g., https://arxiv.org/pdf/1802.06869.pdf) and methods like https://openreview.net/pdf?id=ryj38zWRb which can be seen as symmetric to the proposed one in the sense that an encoderless autoencoder is learnt.

The experimental evaluation is limited. The authors should consider to compare their method with other relevant models such as those mentioned above as well as GANs and their variants. Experiments on other more complex real-world data (e.g., faces) are also needed in order to prove the merits of the proposed model.

---

> ### Author Response · Authors · 2018-11-25
> **Reply to AnonReviewer2**
>
> Thank you for your detailed review!
>
> On the one hand, flow-based models are tractable by design. They allow for data manipulation and their flexibility is mainly limited by the computational resources. However, these models do not allow one to compress the input data.
>
> On the other hand, autoencoders provide a method for significant compression of the data.
> Nevertheless, the training of an autoencoder requires the minimization of a reconstruction error as one of the terms in a total loss function.
> The reconstruction error must be specified beforehand and is dependent on the stated requirements of the task.
>
> When the task is not defined in advance but the compression of the data is required,
> the above-listed methods cannot be used.
>
> The proposed model (PIE) is tractable by design. Moreover, it allows for data compression without specifying the reconstruction error function. The most relevant components are learned from the data.
>
> Thank you for the recommended papers.
> We will consider them during the next revision of the paper.

---

### Official Review · AnonReviewer1 · 2018-11-04

**Rating:** 3
**Confidence:** 4

**Review:**

PIE extend NICE and Real NVP into situations which require having a smaller dimensionality of the latent variable (d) compared to the dimensionality of the observed variable (D), i.e. d < D. This is done by learning an extension function g(z) from R^d to R^{D-d} and then using the change of variables formula on x and [z, g(z)]. To model probabilistically the deterministic function g(z) is replaced by Normal distribution with mean g(z) and a small variance.

PIE is used to build deep generative models and trained on the MNIST dataset. The authors show that the models learnt via PIE produce sharper samples than VAEs and Wasserstein autoencoders (WAEs). No comparison to real NVP is made, which should be the main baseline of comparison to answer the question of "what is the advantage of having d < D?". Further MNIST is no longer a good enough benchmark to evaluate deep generative models. Most representative work in this literature use CIFAR-10, downsampled Imagenet, or Imagenet at 256x256.

This work falls short of the standards of ICLR in a few ways:

1. The presentation is unclear. The explanation of the extension-restriction idea is overly complicated. Further, the paper does very little to properly contextualize this work in the literature. Real NVP and flow-based models are mentioned but the proposed technique is not compared to it. The authors say they "introduce new class of likelihood-based Auto-Encoders", but this is false as far as I understood. The technique is not even an autoencoder since a separate decoder is not trained, and is obtained by exactly inverting the encoder as in real NVP.

2. The experiments are weak. The samples shown are of poor quality, and on a very simplisitic dataset (MNIST). The authors compare with vanilla VAEs, but ignore more recent improvements to VAE such as VAE-IAF, flow-based models, and also autoregressive models. A heuristic is used to measure sharpness and only used to compare against VAE and WAE. Since all these models allow likelihood evaluation, likelihoods should also have been compared.

3. The technique itself is a small change over real NVP and it's not clear whether this change brings any improvements or provides any insights about generative modeling.

---

> ### Author Response · Authors · 2018-11-25
> **Reply to AnonReviewer1**
>
> Thank you for your review!
>
> First of all, we would like to emphasize the fact the PIE is an autoencoder and allows for dimensionality reduction.
> We refer to PIE as an Autoencoder as it performs the encoding of the data automatically.
> In contrast to previously published paper on invertible models, PIE allows for the compression of the data
> and chooses the main nonlinear components of the input.
> The direct comparison to flow-based models such as Real NVP and NICE is not relevant in sense of compression,
> as these models transform the distributions of variables while preserving the dimensionality.
> In section 3.4 we discuss the relation to flow-based models with multiscale architecture
> and demonstrate that such models may be viewed as one of the configurations of PIEs.
>
> In our paper, we start from a necessity of the dimensionality reduction and derive a
> general method for achieving this by using invertible models. The models studied in the experiments are chosen to be simple in order to demonstrate the difference between vanilla methods.
> As always, the proposed model could be used as a backbone for a more complicated setup,
> but it is out of the scope of the current paper.
>
> We agree that the experimental part is limited. We will compare our model to a wider class of competitors during the next revision. Thank you for the recommended models to compare with.

---

### Public Comment · ~Robin_Tibor_Schirrmeister1 · 2018-10-12
**Understanding Questions and related work**

Hi, quite interesting paper!
First, I have some questions for understanding it correctly.

1) In figure 3 c), what is the meaning of the green lines? From the text, I assumed g_k(z_k) should only be computed from those dimensions that will be processed further, correct? So in the case of the second layer, g_k(z_k) only from those 4 dimensions that will be processed further? So why are there green lines from all eight nodes of the second layer to the one node at the bottom? And what should this one node symbolize? r_k? Comparison between r_k and g_k(z_k)?

2) What exactly is used in the generation of data/in the inversion for the values of the r's? In Figure 6 b) it looks as if you put in 0s? Shouldn't you put in g(z)? Or even sample from the gaussian with standard deviation eps_0 centered at g(z)?

Also wanted to mention two possibly related works for your consideration: In our paper "Training Generative Reversible Networks" https://arxiv.org/abs/1806.01610 , we also use a reduced latent dimensionality, however without a rigorous mathematical motivation. In the parallel ICLR submission "Analyzing Inverse Problems with Invertible Neural Network" (https://openreview.net/forum?id=rJed6j0cKX) the authors  experiment with a different kind of partitioning of the latent space.

---

> ### Author Response · Authors · 2018-10-13
> **Reply**
>
> Thank you for your comment!
>
> 1) In Fig. 3 the green lines indicate the aggregation of the variables in objective function.
> For examples, the blue nodes of the 2nd layer which are discarded (r_k)
> and the variables which will be further processed (z_k) are aggregated in a conditional distribution
> p(r_k| z_k) = \delta(r_k - g_k(z_k)).
> We will try to change the scheme in order to avoid confusion.
>
> 2.1) Why 0?
> In experiments with Gaussian PIE we used g(z) = 0.
> Therefore, x = [z, 0] as it is indicated in Eq. 20.
> We will change Fig. 6 (b) in order to exactly match Eq. 20.
>
> 2.2) To sample or not to sample?
> In our experiment we demonstrate the behaviour of function G, defined in Eq. 4.
> The operation of extension of the function is deterministic.
>
> We conducted the experiments, where we used sampling from N(g(z), eps^2 I).
> The obtained images were visually close to those depicted in the current version of the paper.
>
> 3) Thank you for providing us with interesting and useful papers.
> We will consider them during the next revision of the paper.

---

### Public Comment · (anonymous) · 2019-01-13
**Implementation?**


Original bijective net is
F(x)=z
F_inv(z)=x

PIE net is:
F(x)=[z;r] where r~Normal(mu=g(z), sd= <<1 )
F_inv(F(x)) = [z;g(z)]

How do I implement r mapping to Normal?

What is function g(z) to parameterise mu, is it just a Linear layer from d dim to D-d dim?

If g(z) is Linear layer with input_dim =d and output_dim=D-d ,
is the objective to minimize is r_loss = 0.5* -(g(z).mean() - 0.001)).mean()
Thanks in advance

---

> ### Author Response · Authors · 2019-01-13
> **Implementation!**
>
> Hello
>
> >> How do I implement r mapping to Normal?
> In order to train PIE, one maximises the function in Eq. 13
> So r ~ N(g(z), eps^2) if |r - g(z)|_2^2 = 0
>
> >> What is function g(z) to parameterise mu, is it just a Linear layer from d dim to D-d dim?
> g(z) could be any differentiable function from d to D-d
> In our experiments we use g(z) = 0
>
> >> If g(z) is Linear layer with input_dim =d and output_dim=D-d ,
> >> is the objective to minimize is r_loss = 0.5* -(g(z).mean() - 0.001)).mean()
> No. It is r_loss = - ((r - g(z))**2).sum()  / (2 * eps**2)

---

> > ### Public Comment · (anonymous) · 2019-01-13
> > **eps and g(z)?**
> >
> > how do I find
> > eps?

---

> > > ### Author Response · Authors · 2019-01-13
> > > **eps!**
> > >
> > > eps is a scalar hyper-parameter of the method.
> > > Is it introduced in Eq. 10
> > > Its role is discussed after Eq. 17
> > > In experiment 5.1 we demonstrate how it affects the process of the encoding.

---

> > > > ### Public Comment · (anonymous) · 2019-01-13
> > > > **So eps is a fixed constant?**
> > > >
> > > > So eps = constant?
> > > > Also since g(z)=0, is it g(z)=torch.zeros_like(z) or g(z)=z-z
> > > > Thanks again!

---

> > > > > ### Author Response · Authors · 2019-01-13
> > > > > **eps is a fixed constant!**
> > > > >
> > > > > eps is a constant during optimization.
> > > > > dim(g(z)) = dim(r),
> > > > > so g(z) = torch.zeros_like(r)

---

> > > > > > ### Public Comment · (anonymous) · 2019-01-15
> > > > > > **It works!**
> > > > > >
> > > > > > I tested on CIFAR-10, it works like denoising:
> > > > > > In early training:
> > > > > > the recovered (inverted feature) images have black parts (the residual, torch.zeros_like(r), not fully trained)
> > > > > >
> > > > > > In later epochs:
> > > > > > replacing residual output with torch.zeros_like(r), then invert back to image, does image completion (black parts before are filled with some pixels), not 100% accurate, but plausible! hence "pseudo inverse"
> > > > > >
> > > > > > Thank you for your time to explain!

---

> > > > > > > ### Author Response · Authors · 2019-01-15
> > > > > > > **Good!**
> > > > > > >
> > > > > > > You are welcome!

---

> > > > > > > > ### Public Comment · (anonymous) · 2019-01-16
> > > > > > > > **g(z)=0 alternatives?**
> > > > > > > >
> > > > > > > > setting g(z)=0 means PIE is doing denoising variational pseudo inverse
> > > > > > > > is there better alternative other than setting g(z)=0?

---

> > > > > > > > > ### Author Response · Authors · 2019-01-16
> > > > > > > > > **g(z)=0 alternatives!**
> > > > > > > > >
> > > > > > > > > You can choose g(z) to be a neural network or a pre-fixed analytical function.
> > > > > > > > >
> > > > > > > > > By the way, feel free to contact us directly via email

---

### Public Comment · (anonymous) · 2019-09-11
**A general question about invertible auto-encoders**

My question is in general about invertible autoencoders. If we have an invertible autoencoder, does that mean we only need to train the encoder (and not the decoder) since the reconstruction can be obtained by inverting the encoder?

---

> ### Author Response · Authors · 2019-09-12
> **Yes**
>
> If it is invertible, then yes. But the details depend on the exact method you use.

---

### Meta-Review · Area_Chair1 · 2018-12-12
**New approach for an Invertible Architecture Autoencoder shows promise, but experiments incomplete.**

**Confidence:** 3
**Recommendation:** Reject

**Metareview:**

The presented approach demonstrates an invertible architecture for auto-encoding, which demonstrates improvements in performance relative to VAE and WAE's on MNIST.

Pros:
+ R3: The idea of pseudo-inversion is interesting.
+ R3: Manuscript is clear.

Cons:
- R1,2,3: Additional experiments needed on CIFAR, ImageNet, others.
- R1: Presentation unclear. Authors have not made any apparent attempt to improve the clarity of the manuscript, though they make their point that the method allows dimensionality reduction in their response.
- R1, R2: Main advantages not clear.
- R3: Text could be compressed further to allow room for additional experiments.

Reviewers lean reject, and authors have not updated experiments. Authors are encouraged to continue to improve the work.